# Spectral Clustering Identifies High-risk Opioid Tapering Trajectories Associated with Adverse Events

Monika Ray
General Internal Medicine
Center for Healthcare Policy and
Research
University of California Davis Health
United States of America
mray@ucdavis.edu

Joshua J. Fenton
Department of Family and
Community Medicine
Center for Healthcare Policy and
Research
University of California Davis Health
United States of America
jjfenton@ucdavis.edu

Patrick S. Romano
General Internal Medicine
Center for Healthcare Policy and
Research
University of California Davis Health
United States of America
psromano@ucdavis.edu

## ABSTRACT

National opioid prescribing guidelines and related quality measures have stimulated changes in opioid prescribing. Studies have shown that rapid dose tapering may be associated with increased opioid-related and mental health events in some patient groups. However, there isn't enough research on trajectories of dose tapering implemented in clinical practice, and how heterogeneous populations of patients respond to different treatments. Our aim was to examine prescribed opioid doses in a large, longitudinal, clinically diverse, national population of opioid-dependent patients with either Medicare or commercial insurance. We performed phenotype clustering to identify unsuspected, novel patterns in the data. In a longitudinal cohort (2008-2018) of 113,618 patients from the OptumLabs Data Warehouse with 12 consecutive months at a high, stable mean opioid dose (≥50 morphine milligram equivalents), we identified 30,932 patients with one dose tapering phase that began at the first 60-day period with ≥15% reduction in average daily dose across overlapping 60-day windows through seven months of follow-up. We applied spectral clustering as we preferred an assumption-free approach with no apriori information being imposed. Spectral clustering identified several cluster-cohorts, with three that included over 98% of the sample. These three clusters were similar in baseline characteristics, but differed markedly in the magnitude, velocity, duration, and endpoint of tapering. The cluster-cohort characterised by moderately rapid, steady tapering, most often to an end opioid dose of zero, had excess drug-related events, mental health events, and deaths, compared with a cluster characterised by very slow, steady tapering with long-term opioid maintenance. Moderately rapid tapering to discontinuation may be associated with higher risk than slow tapering with longer-term maintenance of opioid analgesia. Furthermore, several clusters highlighted a cohort that had complete taper reversals indicating a treatment failure as the tapering was not maintained. Our findings suggest that identifying subtle yet clinically meaningful patterns in opioid prescribing data, such as patterns within the dose trajectories, can highlight the distinct characteristics separating subpopulations.

## CCS CONCEPTS

• **Applied computing** → **Health informatics**; *Physical sciences and engineering*.

## KEYWORDS

high dose opioids, spectral clustering, patient subpopulations, phenotype clustering, opioid crisis

**ACM Reference Format:**
Monika Ray, Joshua J. Fenton, and Patrick S. Romano. 2023. Spectral Clustering Identifies High-risk Opioid Tapering Trajectories Associated with Adverse Events. In *epiDAMIK 2023: 6th epiDAMIK ACM SIGKDD International Workshop on Epidemiology meets Data Mining and Knowledge Discovery, August 7, 2023, Long Beach, CA, USA*. ACM, New York, NY, USA, 9 pages.

## 1 INTRODUCTION

National prescribing guidelines by the Centers for Disease Control and Prevention (CDC) and the current opioid overdose crisis have led to substantial dose tapering among patients on long-term opioid therapy for chronic pain, especially since 2016 [10, 16, 30]. A quality metric endorsed by the National Quality Forum (NQF) encourages prescribers to reduce opioid doses below 90 morphine milligram equivalents (MME) per day [33]. In the setting of long-term opioid therapy for chronic pain, several studies have shown worse outcomes associated with rapid dose reduction [1, 13, 17, 41] and dose tapering has emerged as a complex issue for both physicians and patients. To better inform evidence-based clinical practices, health system policies, and public programmes, it is necessary to characterise population heterogeneity (phenotype clustering) and to understand which patients are appropriate candidates for different tapering approaches. This type of research requires a better understanding of the variety of tapering trajectories that clinicians implement in diverse populations to enable comparisons of the risks and benefits of alternative approaches in relevant subpopulations. Large healthcare data warehouses that accumulate longitudinal records from multiple sources offer great opportunities for improved understanding of population heterogeneity in opioid dose management.

To undertake this research, we used retrospective data from the OptumLabs Data Warehouse (OLDW), which includes longitudinal health information for over 109 million commercial enrollees and 12.5 million Medicare Advantage enrollees. We leveraged the retrospective cohort previously created by Agnoli and colleagues [1], whose prior research suggested that the peak tapering velocity has

a significant mean effect on adverse outcomes. However, opioid-dependent patients with chronic pain often resist any dose reduction, while pharmacies and regulators encourage dose reduction for every eligible patient. To inform better clinical practice and policies, we need to understand how the peak tapering velocity fits into overall patterns of opioid dose management over time, and then explore the characteristics of higher- and lower-risk subpopulations of patients undergoing dose tapering. For this purpose, we used spectral clustering to describe clinically meaningful subpopulations. Specifically, we wanted to examine similarities among patients within a cluster and differences among patients across clusters. Spectral clustering has been applied to speech processing, computer vision and exploratory data mining in biology [3, 6, 11, 21, 38, 42], but opioid dosing is a novel and highly topical application in the current era of increasing opioid-related overdose death rates [15].

This work deviates from the popular hypothesis-driven approaches where the functional form of the models are independent predictors and dependent outcomes. In this data-driven approach the aim is to first cluster phenotypes, without classifying features as independent or dependent variables, and then identify meaningful signatures within these clusters [25]. These signatures can then be used in predictive models as either predictors or outcomes. The main purpose of phenotype clustering is to uncover hidden patterns. The primary focus of our exploratory work is see (1) how the patients cluster based on their phenotypes (grouping patterns or phenotypes) and (2) whether these clusters have any remarkable differences (i.e., identify signatures that can be used in predictive analytics).

## 1.1 Data Cohort and Adverse Events

We obtained data from 2008-2018 for adults from the OptumLabs Data Warehouse (OLDW) which contains de-identified administrative claims data, including medical and pharmacy claims and eligibility information for commercial and Medicare Advantage enrollees, representing a mixture of ages and regions across the United States. The entire cohort, which we received from Agnoli and colleagues [1], had a stable baseline period of 12 consecutive months at a high opioid dose $\geq$50 MME, resulting in 113,618 patients. The tapered cohort was defined as the subset of patients who had a dose tapering phase, which began on the first 60-day period with $\geq$15% reduction in average daily dose across overlapping 60-day windows through the initial seven months of follow-up. Patients who had $\geq$15% reduction in average daily dose over a longer time frame were not included due to uncertainty about the intent of slight MME dose reductions (which could be driven by delays in picking up prescriptions). To facilitate interpretation we selected a population of patients who had only one period of tapering. Mortality in the tapered cohort was determined by analysing the time after taper initiation and matching against the records in the OLDW mortality table.

Adverse events included emergency department (ED) visits or hospitalisations for (1) drug or alcohol overdose or withdrawal (drug-related events); and (2) depression, anxiety, or suicide attempts (mental health events). Drug-related and mental health events were identified using International Classification of Diseases, Tenth Revision, Clinical Modification (ICD-10-CM) diagnosis codes

for claims from October 2015 through 2019 and ICD-9-CM diagnosis codes for claims from 2008 through September 2015. Comorbidities were identified for all patients using the available software (AHRQ "Elixhauser" Comorbidity Software) in the OLDW [12, 29]. This project was determined by the University of California Office of the President to be exempt from human subjects review, as the OLDW uses completely de-identified, anonymised data.

## 1.2 Analytic Methods

We considered several methods to identify subpopulations and their characteristics such as $K-Means$ clustering and latent class analysis (LCA). $K-Means$ clustering is a popular clustering algorithm but it is based on many restrictive assumptions, which most real-world datasets violate [20, 35]. The algorithm operates on the input data matrix and, hence, is sensitive to the size of the data ($N$) as well as number of features. LCA [23, 43], a type of finite mixture model, may be suitable for describing dose trajectories, but it requires an outcome to be specified. By comparison, spectral clustering is purely unsupervised and does not require outcome variables. For our analyses, we used a novel spectral clustering algorithm (Spectrum) developed by John and colleagues [21]. Spectral graph theory associates the spectrum of a matrix, i.e. eigenvalues of a matrix, to the properties of a graph via the Laplacian matrix [7, 8, 37]. It operates on graphs that are constructed between neighbouring nodes that represent data points (i.e., patients). It identifies arbitrarily shaped clusters (with convex or non-convex boundaries) using the eigenvectors in the Laplacian similarity matrix [7, 9, 26, 46]. A Laplacian similarity matrix models the local neighborhood relationships between data points as an undirected graph [4, 37, 40]. Spectral clustering is robust to the geometry of the clusters and outliers, and does not require the user to specify the number of clusters [2, 24, 46]. It identifies the number of clusters by computing the differences between the consecutive ordered eigenvalues of the graph Laplacian and identifying the first pair of consecutive eigenvalues with the maximum difference in their values.

The steps of spectral clustering include - (1) creation of the similarity matrix, then (2) the creation of the Laplacian matrix, and finally (3) creation of clusters [32, 44]. Variations of spectral clustering algorithms address issues related to creation of the similarity matrix, graph-partitioning and speed on massive datasets. Since spectral clustering operates on the Laplacian similarity matrix, which is an $N$ x $N$ matrix of $N$ data points, it is sensitive to the size of the data. The Spectrum algorithm developed by John et al., is novel in the way it combines the following features - (1) combined Zelnik-Manor self-tuning [49], and the Zhang density-aware [50] kernels to create the similarity matrix, (2) Ng spectral clustering method to estimate the optimal number of clusters [31], and Gaussian mixture modelling (GMM) [47] to finally cluster the data, and (3) a fast approximate spectral clustering (FASP) method [48] to allow for fast clustering of massive data on regular desktop machines. The self-tuning component of the kernel adjusts to the scale of the data, while the density-aware component adapts to the local density of the data creating more or fewer connections depending on the density of the regions. Spectrum uses the diffusion of tensor product graphs (TPG) to capture higher order information in the data and highlight underlying patterns in the data [39]. The final

clusters are plotted using the first two principal components, PC1 and PC2. We did not use the eigen gap-statistic to determine the number of clusters as it was not essential for us to constrain the number of clusters nor were we against identifying small cohorts if the cohort had important patterns to investigate further. In our work, we were searching for anomalies or 'interesting patterns' that could explain the underlying population heterogeneity. The eigen gap heuristic works well if there are well-defined clusters but not of much help when there are noisy or overlapping clusters, which is likely to be the case in this data.

The variables in the input space of the spectral clustering algorithm were age, gender, monthly average opioid dose (MME), mean baseline dose, count of drug-related events in the pre-taper and after tapering initiation phases, the number of mental health events in the pre-taper and after tapering initiation phases, benzodiazepines co-prescription at baseline and at 30 days, 31 Elixhauser comorbidity flags, and the change in dose across consecutive months for 12 months. The number of drug-related and mental health events were identified for each patient before taper and after taper initiation as these were the adverse events of interest. We reviewed each cluster to identify the prevalence of different adverse events as well as the number of deaths after taper initiation. We report the distinguishing characteristics across the cluster subpopulations. For counterfactual inference, we identified the number and proportion of drug-related and mental health events in each cluster, and then computed the excess number of those events relative to the null assumption of equal event risk across all clusters. The counterfactual calculation for each adverse event is given by - $ExcessEvents = (NumEventsCluster) - (NumPatientsCluster * (\frac{TotalEvents}{TotalPatients}))$, where, for each adverse event, i.e., mortality, drug-related events or mental health events, ExcessEvents is the number of excess events in the cluster, NumEventsCluster is the number of observed events within the cluster, NumPatientsCluster is the number of patients in the cluster, TotalEvents is the total number of adverse events in the entire data and TotalPatients is the total number of patients in the analysis.

## 2 RESULTS

Among the 113,618 patients in the entire cohort 33,628 had one or more phases of opioid dose tapering (29.5%) based on the tapering definition of ≥15% reduction in average daily dose in 7-months of follow-up [1]. Fig. 1 shows the analytical pipeline and the resultant plot of the 10 clusters identified. We could not show all the ten clusters clearly in a 2-D plot. Since spectral clustering plots the clusters by collapsing them onto the first two principal components, the multi-dimensional aspect of the clusters is not visible. However, Fig. 1 shows that the clusters are not spherical and the data has outliers. Table 1 shows the characteristics of patients who tapered; the sample was 54% female and 92% had only one tapering period available for analysis.

Spectral clustering of 30,932 patients who underwent single tapers resulted in 10 clusters (groups of patients or subpopulations) with relatively similar baseline characteristics. All clusters had patients with high mean baseline doses of 140-237 MME/day. Of particular interest were the three large clusters and their baseline

characteristics shown in Table 2. The other seven clusters' characteristics are discussed below but not shown due to small cell size policy. The three large clusters (1, 2, and 10) were very similar demographically, with mean ages of 58.7, 57.0, and 58.4 years, and 56%, 53%, and 50% female composition, respectively. They were also similar on baseline co-prescribing of benzodiazepines (29%, 30%, and 30%, respectively) and comorbid diagnoses during the baseline year, such as alcohol abuse and dependence (2%, 3%, and 2%, respectively), drug abuse and dependence (17%, 17%, and 15%, respectively), and depression (32%, 31%, and 30%, respectively). Furthermore, they had similar medical experiences during their pre-taper period of stable opioid dosing, with relatively few drug-related events (mean 0.042, 0.053, and 0.043, respectively) and more mental health events (mean 3.81, 4.03, and 3.66, respectively).

Fig. 2 compares the tapering trajectories across clusters. Each trajectory is plotted as the average monthly dose of the patients in the cluster. The three largest clusters had markedly different opioid dose tapering trajectories and associated adverse events as shown in Table 3. The number of excess events represents the difference between the number of observed events and the number of events that would have occurred if all the clusters had the same event rate. About 55% of patients were in cluster 1, characterised by very slow and steady tapering to a final dose about two-thirds of baseline, with low event rates and no reversal to pre-taper baseline dose. While clusters 2 and 10 looked quite similar in their baseline characteristics, they had very different taper trajectories. Cluster 2 was characterised by relatively rapid tapering to zero or very low doses, while cluster 10 was characterised by somewhat slower tapering from lower baseline doses to higher end doses. Both these clusters had slightly higher event rates than other clusters. Clusters 2 and 10 also had more drug-related events than cluster 1 (mean 0.116 and 0.128 versus 0.074), more mental health events (mean 0.089 and 0.075 versus 0.058), and more deaths (mean 0.079 and 0.098 versus 0.036) during the tapering year. However, compared to cluster 10, cluster 2 had higher baseline mean and median doses (192.3 and 137.0 MME versus 140.3 and 104.0 MME), and a lower mean end dose (12.9 versus 37.6 MME). The slow trajectory for cluster 1, and the very low or zero doses in clusters 2 and 10, continued into the 15th month, although those months were not included in the spectral clustering analyses.

The characteristics of the taper trajectories for all the clusters are detailed in Table 4. The left panel in Fig. 3 shows the proportion of patients with 0 MME dose of opioids across the three clusters each month, while the right panel shows the taper trajectory. Table 5 shows the relative change in the proportion of patients who were prescribed 0 MME opioids at each time point in the three clusters. Cluster 2 had the highest proportion of patients (73%) who were completely tapered off opioids at the end of 12 months, compared to cluster 10 (66%) and cluster 1 (2%). Since cluster 1 demonstrated the safest outcomes, we compared clusters 2 and 10 to cluster 1. The graph in the left panel in Fig. 3 shows that cluster 2 had a steep yet steady upward trend in the proportion of patients who were taken off opioids, whereas patients in cluster 1 almost uniformly stayed on opioids, and cluster 10 demonstrated a pattern of delayed discontinuation.

The remaining 1.3% of patients sorted into seven smaller clusters, all of which had patients who were tapered to or close to 0 MME

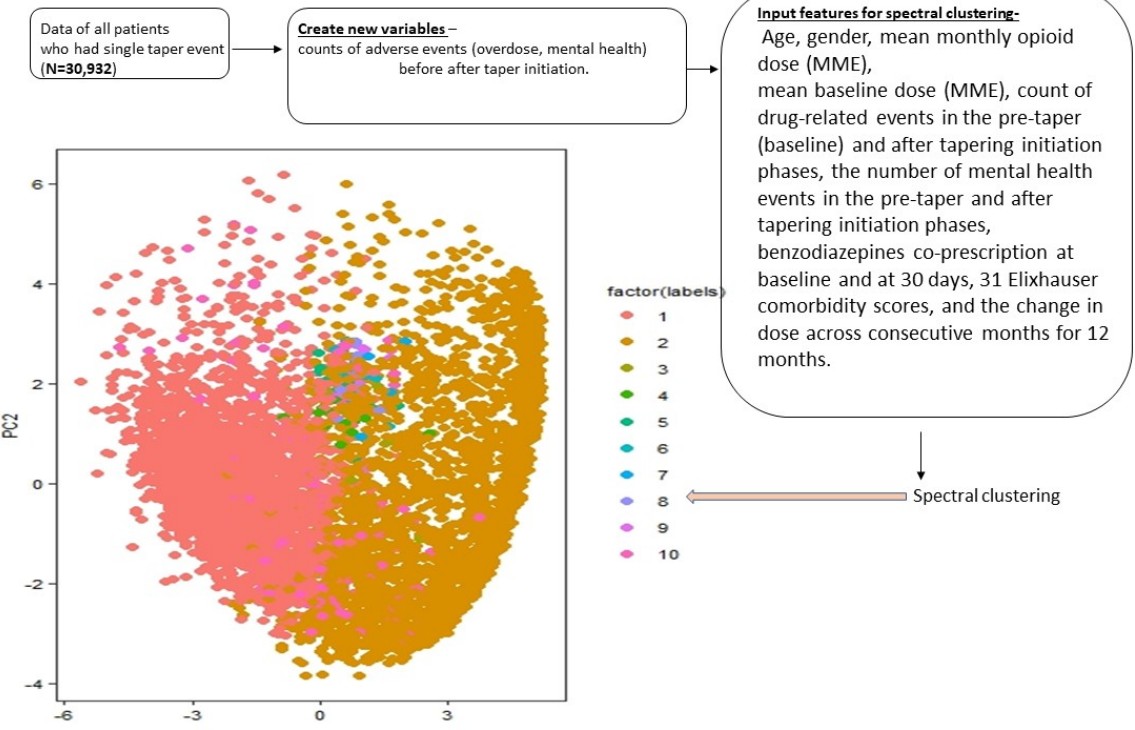

**Figure 1: Analysis Flowchart**

**Table 1: Characteristics of the patients who tapered**

| Variables | Categories | n |
|---|---|---|
| Gender | Female | 18,197 |
| | Male | 15,431 |
| Age | Mean±Std. | 58.0±11.6 |
| Number of Tapers | 1 | 30,932 |
| | 2 | 2,462 |
| | >= 3 | 234 |
| Number of drug-related events before tapering | 0 | 32,238 |
| | 1 | 1,182 |
| | >= 2 | 208 |
| Number of drug-related events after tapering | 0 | 31,210 |
| | 1 | 1,888 |
| | 2 | 356 |
| | >= 3 | 174 |
| Number of mental health events before tapering | 0 | 14,788 |
| | 1 | 3,984 |
| | 2 | 2,949 |
| | 3 | 2,040 |
| | 4 | 1,665 |
| | 5 | 1,223 |
| | 6 | 1,034 |
| | >= 7 | 5,945 |
| Number of mental health events after tapering | 0 | 32,041 |
| | 1 | 1,096 |
| | 2 | 300 |
| | >= 3 | 191 |

**Table 2: Characteristics of Clusters 1, 2 and 10 in the pre-taper period**

| Cluster | No. patients | Age (Mean) | Female (%) | benzodiazepines Rx (%) | Alcohol abuse (%) | Depression (%) | Drug abuse (%) | Drug-related event counts (Mean) | Mental Health event counts(Mean) | Base dose (Mean MME) |
|---|---|---|---|---|---|---|---|---|---|---|
| 1 | 16,965 | 58.74 | 55.7 | 28.9 | 2.4 | 31.7 | 16.6 | 0.04 | 3.81 | 189.82 |
| 2 | 13,025 | 56.96 | 53.1 | 30.1 | 3.0 | 31.4 | 16.5 | 0.05 | 4.03 | 192.31 |
| 10 | 531 | 58.36 | 49.5 | 29.7 | 3.4 | 30.3 | 15.1 | 0.04 | 3.66 | 140.33 |

**Table 3: Adverse events after taper initiation in clusters 1, 2 and 10**

| Cluster | No. patients (%) | Drug-related events/1000 | No. Excess drug-related events | Mental Health events/1000 | No. Excess Mental Health events | Deaths/1000 | No. Excess Deaths |
|---|---|---|---|---|---|---|---|
| 1 | 16,965 (55%) | 74.0 | -320.2 | 58.4 | -240.2 | 36.1 | -329.8 |
| 2 | 13,025 (42%) | 116.2 | 303.6 | 89.4 | 220.5 | 79.1 | 306.2 |
| 10 | 531 (< 2%) | 128.1 | 18.7 | 75.3 | 1.5 | 97.9 | 22.5 |

**Table 4: Average monthly dose for 12 months from taper initiation - Taper Trajectories**

| Cluster | BaseDose | Mon1 | Mon2 | Mon3 | Mon4 | Mon5 | Mon6 | Mon7 | Mon8 | Mon9 | Mon10 | Mon11 | Mon12 | Taper Trajectory |
|---|---|---|---|---|---|---|---|---|---|---|---|---|---|---|
| 1 | 189.82 | 174.53 | 170.27 | 165.64 | 161.23 | 157.28 | 154.15 | 155.05 | 155.53 | 155.25 | 154.05 | 151.68 | 144.01 | Very slow, no reversal |
| 2 | 192.31 | 175.19 | 157.04 | 139.42 | 119.01 | 96.06 | 75.19 | 59.71 | 45.49 | 33.53 | 23.35 | 15.18 | 12.90 | Rapid, no reversal |
| 3 | 236.81 | 213.18 | 121.69 | 1.38 | 193.46 | 204.26 | 206.02 | 191.60 | 163.58 | 150.98 | 141.49 | 129.90 | 114.59 | Very Rapid, complete reversal |
| 4 | 192.57 | 179.16 | 0.44 | 185.31 | 194.26 | 194.64 | 176.29 | 167.38 | 160.98 | 150.52 | 143.25 | 134.76 | 133.31 | Very Rapid, complete reversal |
| 5 | 196.99 | 183.05 | 147.09 | 92.71 | 0.33 | 172.22 | 176.60 | 158.29 | 145.41 | 139.10 | 135.23 | 119.75 | 113.12 | Very Rapid, complete reversal |
| 6 | 212.81 | 205.10 | 182.34 | 153.96 | 106.37 | 77.02 | 5.26 | 0.00 | 168.49 | 169.27 | 152.98 | 120.84 | 115.09 | Very Rapid, complete reversal |
| 7 | 227.55 | 217.24 | 171.99 | 152.88 | 122.05 | 101.76 | 57.73 | 31.72 | 22.56 | 0.00 | 148.42 | 147.73 | 135.03 | Rapid, partial reversal |
| 8 | 217.07 | 205.71 | 177.62 | 161.43 | 145.93 | 102.60 | 78.04 | 64.87 | 51.06 | 33.13 | 0.00 | 157.58 | 166.52 | Rapid, partial reversal |
| 9 | 220.37 | 203.30 | 160.72 | 117.39 | 85.31 | 63.20 | 59.18 | 48.60 | 36.30 | 29.20 | 18.94 | 0.00 | 143.26 | Rapid, partial reversal |
| 10 | 140.33 | 124.30 | 114.04 | 111.72 | 109.34 | 101.91 | 92.57 | 85.40 | 80.46 | 100.04 | 101.61 | 81.17 | 37.57 | Erratic, no reversal |

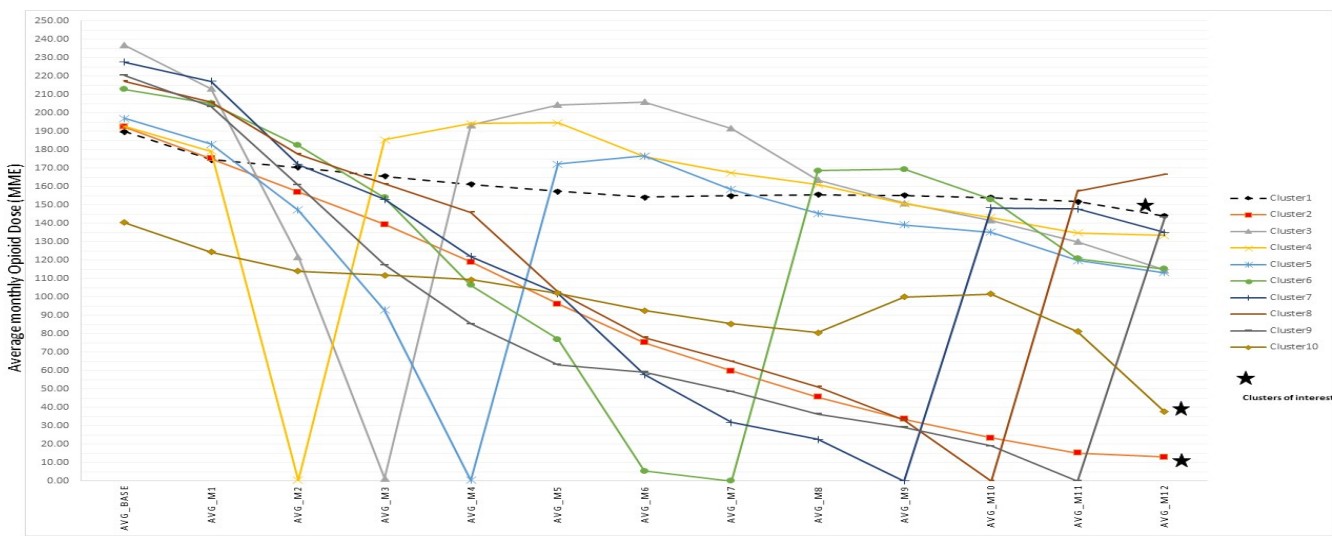

**Figure 2: The average monthly dose in MME for all the patients within each cluster.**

(not shown due to small cell size policy). In clusters 3, 4, and 5, dose tapering to near zero occurred very rapidly within 4 months after initiation, but the pre-taper dose was quickly restored and slow tapering was initiated instead. On the other hand, in clusters 6, 7, 8, and 9, rapid tapering occurred over a longer period of 6-11 months, but the taper was largely reversed and the subsequent trajectory was truncated due to the cohort design. Drug-related event rates

and mental health event rates were quite variable across these small clusters (data not shown), but in aggregate, the mental health event rate of patients in these seven clusters was over twice that of cluster 1 (mean 0.117 versus 0.058).

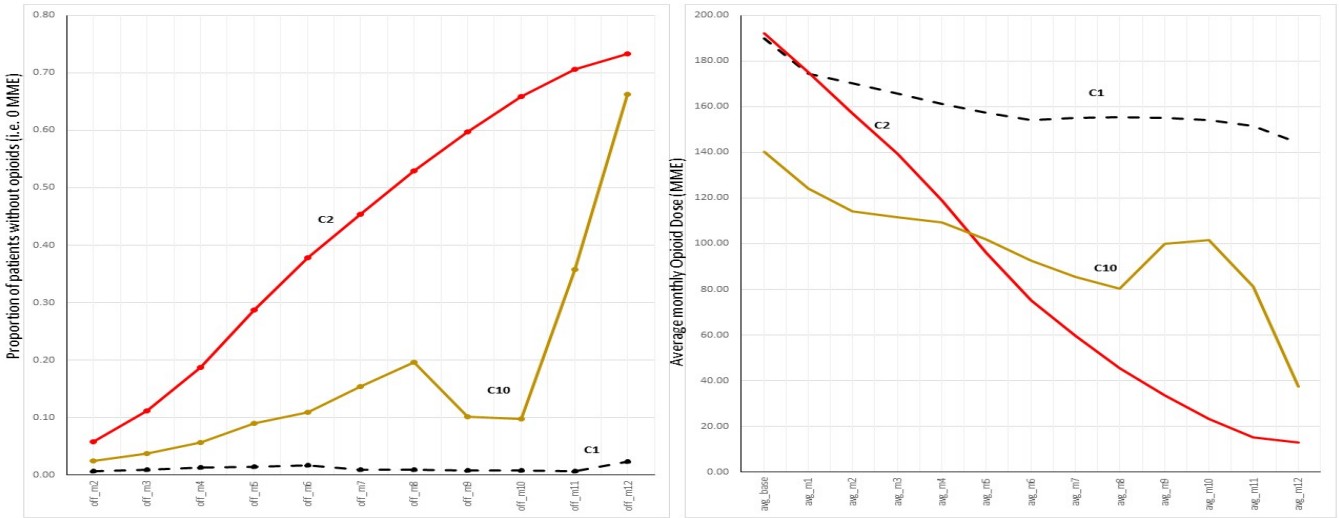

**Figure 3: The proportion of patients without opioids, i.e., with an average monthly dose of 0 MME, in the three clusters of interest and their corresponding tapering trajectories.**

**Table 5: Relative change in the proportion of patients who were prescribed 0 MME opioids by month**

| Month | C1 Prop. Patients | C1 Relative change | C2 Prop. Patients | C2 Relative change | Diff.Relative changes C1 - C2 | C10 Prop. Patients | C10 Relative change | Diff. Relative changes C1 - C10 |
|---|---|---|---|---|---|---|---|---|
| 2nd | 0.007 | | 0.058 | | | 0.024 | | |
| 3rd | 0.010 | 0.046 | 0.112 | 0.95 | -0.49 | 0.038 | 0.54 | -0.08 |
| 4th | 0.013 | -0.99 | 0.187 | 0.66 | -1.65 | 0.056 | 0.50 | -1.49 |
| 5th | 0.015 | 0.13 | 0.287 | 0.54 | -0.41 | 0.090 | 0.60 | -0.47 |
| 6th | 0.016 | -0.98 | 0.378 | 0.32 | -1.30 | 0.109 | 0.21 | -1.19 |
| 7th | 0.009 | -0.46 | 0.454 | 0.20 | -0.66 | 0.154 | 0.41 | -0.87 |
| 8th | 0.010 | -0.99 | 0.530 | 0.17 | -1.16 | 0.196 | 0.27 | -1.26 |
| 9th | 0.008 | -0.21 | 0.597 | 0.13 | -0.34 | 0.102 | -0.48 | 0.27 |
| 10th | 0.008 | -0.99 | 0.659 | 0.10 | -1.10 | 0.098 | -0.04 | -0.95 |
| 11th | 0.007 | -0.15 | 0.707 | 0.07 | -0.22 | 0.358 | 2.65 | -2.80 |
| 12th | 0.024 | -0.98 | 0.733 | 0.04 | -1.01 | 0.663 | 0.85 | -1.83 |

Relative change refers to the difference in the proportion of patients within the cluster between the current and the previous month. Negative value indicates that fewer patients were prescribed 0 MME opioid in the current month compared to the previous month. C1- Cluster 1; C2- Cluster 2; C10- Cluster 10.

## 3  DISCUSSION

In this large longitudinal cohort of patients with chronic pain receiving high dose opioids at stable dosing for at least one year, spectral clustering analysis suggested wide variability in dose tapering patterns over the first year of tapering. These trajectories show notable variation in the velocity and duration of tapering, post-tapering minimum doses and subsequent re-initiation (taper reversal) of moderate-to-high opioid doses, which was an unexpected finding. While the specific number of clusters is not important, the cohorts identified were interesting and are discussed here. The largest cluster (cluster 1 with 55% of patients) was characterised by very slow, gradual tapering from a mean baseline dose of 190 MME to 144 MME at 12 months, whereas the second largest cluster (cluster 2

with 42% of patients) was characterised by quicker and steep tapering from a mean baseline dose of 192 MME to only 12.9 MME (with 73% of patients discontinued). The latter cluster, unlike other clusters, had a substantial excess of both drug-related and mental health events after the initiation of tapering, suggesting that tapering patients accustomed to high-dose prescription opioids to zero may be associated with important health risks. Our results suggest that there is a significant subpopulation of patients receiving high-dose opioids for chronic pain who may not tolerate tapering to very low doses. Many of these patients may have had opioid use disorders; previous research in the OLDW has shown that such patients have better outcomes if treated with buprenorphine or methadone [45].

There wasn't any strong rationale to specify the number of clusters as we were looking for 'interesting patterns' which could seem

like outliers compared to the rest of the data. Notably, spectral clustering identified previously unsuspected and unusual patterns in the opioid dose management data. In particular, two small clusters were characterised by rapid tapering to negligible or zero doses, followed by re-initiation of prescription opioids at moderately high doses. These patterns merit further exploration as they strongly suggest that reversal of tapering may be a marker of an unsuccessful tapering strategy and that clinicians can safely resume prior opioid doses for some of these patients. These patients with unsuccessful tapers need to be separated and studied alongside the group of successful tapers rather than be combined as was done when this cohort was selected for analysis (See Data Cohort and Adverse Events section). This suggests that the definition of a tapered cohort needs to be re-visited and taper reversals be counted as an adverse event. Our findings highlight the importance of considering the velocity of tapering, as suggested by Agnoli and colleagues' research, along with the taper duration and post-tapering final dose as clinicians attempt to devise safer dose tapering strategies to address the current opioid overdose epidemic in the US. Unsupervised data mining methods are powerful tools when the aim is to understand the data better and see what may have been previously missed in hypothesis-driven studies. Lastly, unsupervised knowledge discovery research helps in extracting novel, unsuspected phenomena that can be investigated using supervised methods. These methods may also challenge what was previously thought to be true; for example, by identifying previously unrecognised patterns of tapering reversal shown in Fig. 2.

During the writing of this manuscript, another report was published that analysed trajectories in patients receiving long-term opioid therapy using based trajectory modeling (GBTM) [5]. Binswanger's analysis identified five trajectories. From the clinical perspective, this is interesting but is an oversimplification as it puts all tapering patients into two groups – one slightly decreasing (which they reassigned to the stable group) and one decreasing (which they compared with the stable group) but they did not clearly identify taper reversals, suggesting that all tapers are maintained over time. We selected our cohort based on whether they tapered at some point but did not filter to select those with decreasing trajectories based on different velocities. Hence, it is quite plausible to expect multiple groups. In addition to being fully exploratory, with no assumptions on what kind of trajectories to expect, our analysis focused on patients for whom a taper was pre-determined to understand the different types and speeds of tapering. Therefore, our results support and facilitate future analyses comparing the outcomes of these different tapering approaches with the alternative of not tapering at all (a control group of non-tapers), which is a viable approach but was not represented in our sample. Other notable difference from Binswanger's work is that we did not assume any data properties such as distributions, number of anticipated clusters, etc. to run spectral clustering and our dataset is many times larger and representative of the entire population in the US. As we were searching for subtle differences in a population that consists of tapering patients, in order to receive an amplified signal, we need a large cohort and use methods that do not impose any assumptions on the input data or the results. This is exactly what knowledge discovery is, i.e., where the scholar keeps an open mind about the kind of patterns/information that will emerge. Unlike Binswanger's

report, we did not impose any restriction on the spectral clustering algorithm. It was during the analysis of clusters to understand why the patients segregated as such, did we notice that the *pattern of the trajectories were the point of subtle difference* and discussed this in detail. This is work in progress as we will need to further analyse these patterns using parametric methods and also study other potential outcomes of such tapering patterns. For the purpose of knowledge discovery with no apriori information, we preferred an assumption-free approach with no apriori information being imposed in any phase of the analysis. Furthermore, as we did not have any prior knowledge of the underlying distribution patterns in this cohort, GBTM could have led us to incorrect results [28]. GBTM relies heavily on prior information which, in essence, is a different approach than the one here which was to identify patterns that automatically emerge and would correlate with nuanced differences in an already tapering population.

We acknowledge some limitations in our analyses such as unknown intent of the prescribing provider. For example, the physician's choice of a rapid or slow taper may be driven by unobserved characteristics of patients or their medical histories, which may independently contribute to the resulting outcomes. We were also unable to distinguish patient-supported tapering from physician-demanded tapering and what may have triggered taper reversals. Finally, the current data do not capture illicit opioid use, sharing of opioids prescribed for other patients, or methadone administered in certified treatment programmes. Nevertheless, our study is relevant to the research and clinical communities grappling with the opioid crisis. There is substantial interest in understanding factors contributing to the current epidemic of opioid-related overdose deaths [15], reflected in several recent economic analyses on physician prescribing patterns and opioid abuse [18, 22], statewide surveys and reports on prescribing practices and patient outcomes [14, 27, 34], and studies of physician prescribing patterns and outcomes [19, 36]. Previous studies of opioid dose tapering either used smaller, less nationally representative cohorts or relied on supervised analytic methods, where an outcome is always defined, to identify patient characteristics that are associated with adverse outcomes.

## 4 CONCLUSION

Our objective was knowledge discovery, which was to identify hidden, unsuspected patterns in claims data for patients with chronic pain. Since our analysis was performed using a large dataset that is representative of the population of the United States these results are generalisable. The insights from this work will be used to extend this work and guide predictive analysis. Our study also highlights the need for more detailed investigations to identify what patient factors should be considered while suggesting a dose tapering regimen. Dose tapering to discontinuation may plausibly increase the risk of subsequent opioid overdose if these opioid-dependent patients seek alternative opioids from illicit sources or mix opioids with other sedating drugs such as benzodiazepines, thereby negating the purpose of dose tapering. We find these results, obtained using a data driven approach, to be compelling enough to warrant further investigations into dose tapering patterns to inform future national prescribing policies and clinical practice.

## ACKNOWLEDGMENTS

The authors extend their sincere gratitude to Guibo Xing, Elizabeth Magnan, Alicia Agnoli and Daniel Tancredi for data sharing as well as members of the OptumLabs OLDW team for their valuable guidance.

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
