# OpenReview forum: "Spectral Clustering Identifies High-risk Opioid Tapering Trajectories Associated with Adverse Events"
_KDD.org/2023/Workshop/epiDAMIK — KDD 2023 Workshop epiDAMIK_

### Official Review · Reviewer_hQpY · 2023-06-16
**Accept**

**Rating:** 4
**Confidence:** 4

**Review:**

In this study, the authors applied spectral clustering to identify high-risk opioid tapering trajectories associated with adverse outcomes using a large-scale dataset.  The study addressed an important public health issue and discovered patterns of opioid tapering using unsupervised learning. The findings can support further studies on dose tapering patterns to inform future prescribing policies and clinical practice. A couple of areas can be improved in the current manuscript.
1. More details should be provided on creating the similarity matrix and the Laplacian matrix. A few references were mentioned but without technical details.
2. Variables in this study have different units and scales. How did the author deal with the different scales? Did it matter if those variables were standardized?
3. Certain patients may use dose tapering strategies based on their health conditions. For instance, slower tapering may be used because the doctor believes the patient may have a higher risk of adverse outcomes under rapid tapering. In future studies, it would be better to control those factors to disentangle the impact of tapering patterns. A discussion on this point is warranted.

---

### Official Review · Reviewer_f7RV · 2023-06-27
**Spectral Clustering Identifies High-risk Opioid Tapering Trajectories Associated with Adverse Outcomes- Review**

**Rating:** 3
**Confidence:** 3

**Review:**

**Summary:**
This work explores the characteristics of the prescribed opioid doses in a diverse population of opioid-dependant patients with appropriate insurance and employer of licensed physicians. Namely, the work applies different clustering methods to identify patterns in the data. The dataset used in this work was from the claims data about patients with chronic pain obtained from the largest commercial insurance company and the largest private employer of physicians in the United States. The authors justify that spectral clustering might be a more suitable unsupervised claustering algorithm compared to other clustering methods and thus use that to construct clusters. The authors then explored the characteristics of the patients present in each of these clusters by defining counterfactual terms and other evaluation metrics. Notably, the work discovered different observations related to each of the clusters.

**Strong Points:**
+ The work provides valuable insights into the characteristics related to patient behaviour towards opioid prescriptions.
+ The fact that the authors closely describe the remotely relevant prior study (GBTM) and accurately describe the distinctions between that study and the current work attest to the specific contribution of this work.
+ The work provides extensive summaries of patients across the different clusters over pre-taper and taper initiation and post-taper timestamps. This provides important insights about characterizing patient dose trajectories.

**Weak Points:**
+ The spectral clustering algorithm used in this work (Spectrum) is a readily available R package. As the authors do not necessarily create this algorithm, it seems pointless to spend an entire subsection on the given algorithm as the characteristic comparison with other clustering algorithms is not unique as well.
+ I am not sure if the number of clusters given in this work is just like making a mountain out of a molehill. This is because based on Figure 1, there seems to mainly be 2-3 broad clusters. Although the authors provide intuitions for the other clusters, it seems that clusters 3,4 and 5 have very similar trends based on Figure 2. It may be more informative if the authors provided the values of the Eigen-Gap statistic for each value of K where K denotes the number of clusters. Maybe GBTM is not as much of an oversimplification as the authors make it to be.


**Minor Suggestions:**
+ Figure 1 needs to be broken down into 2 separate plots. 1 plot for the 3 main clusters and the other for the other clusters. Otherwise the smaller clusters are not even visible.
+ Minor grammatical errors seem to be present, like in line 737, it should probable be "Finally, the current data *does* not ...".

---

### Official Review · Reviewer_G1cy · 2023-06-30
**An interesting approach that may require more supporting analysis**

**Rating:** 3
**Confidence:** 5

**Review:**

In this paper, the authors studied the problem of identifying meaningful clinical patterns among patients who had been prescribed opioids and subsequently had the doses reduced over different lengths of time. Overall the paper has several strong aspects
- It was able to identify several dosage patterns that maybe of interest towards clinical determination
- The initial analysis seems to point towards differing health outcomes for patients with slow vs rapid tapering (see more below)
- The paper covered sufficient details about cohort characteristics to let the reviewers judge the impact of the findings

However, from a health economic outcome research aspect, the paper is currently at an early stage and may need further followups to support the validity of the identified patterns. The authors have acknowledged the limitation of not considering other factors that may capture the intent to reduce/increase dosing. However, this is a key aspect that may need to be validated, perhaps with certain assumptions such as IPW, to satisfy the significance of the findings.  Further, the authors may want to considering survival analysis methods, especially with the possibility of right censored events, to further analyze the clinical outcomes of the identified cohorts.

Overall, this papers has certain promises but may be improved upon from a modeling and analysis aspect.

---

### Official Review · Reviewer_beJQ · 2023-06-30
**Review of paper 1**

**Rating:** 3
**Confidence:** 4

**Review:**

This paper aims to study tapering trajectories for patients on long-term opioid therapy. They use longitudinal health data from United HealthGroup and identify 33,620 patients who underwent dose tapering. They apply spectral clustering (a variant by John et al., 2019) to cluster the patients, using variables including their age, gender, monthly average opioid dose, mean baseline dose, tapering trajectory, and adverse events pre-tapering and after tapering initiation. They find 10 clusters and focus on the three largest ones, which exhibit different tapering trajectories and slightly different adverse outcomes, while looking mostly similar on baseline characteristics.

Strengths
+ The problem is an important one to understand, i.e., the risks and benefits of dose tapering
+ The dataset is strong and appropriate for the study - they are able to identify over 33,000 patients with dose tapering and longitudinal data
+ It is interesting that there are different tapering trajectories discovered

Weaknesses/suggestions
- The goal of the work seems to be 1) to identify common tapering trajectories, 2) to learn the relationship between those trajectories and adverse outcomes. It's not clear to me, then, why clustering on all the variables – the baseline characteristics, the dose trajectory, and adverse events – is the right method here. Instead, would it make more sense to do something like, only cluster on dose trajectory, in order to answer (1), and then to fit a model to say, controlling for baseline characteristics, what is the effect of this kind of trajectory on adverse events, in order to answer (2)?
- The comparison of adverse events across clusters is also difficult to interpret without confidence intervals (Table 3)